# Whole-Transcriptome Analysis Reveals the Regulatory Network of Immune Response in Dapulian Pig

**DOI:** 10.3390/ani14233546

**Published:** 2024-12-08

**Authors:** Tao Wang, Zhe Tian, Mubin Yu, Shuer Zhang, Min Zhang, Xiangwei Zhai, Wei Shen, Junjie Wang

**Affiliations:** 1College of Life Sciences, Key Laboratory of Animal Reproduction and Biotechnology in Universities of Shandong, Qingdao Agricultural University, Qingdao 266109, China; wangtao990703@163.com (T.W.); tianzhe0609@163.com (Z.T.); yumubin0402@163.com (M.Y.); 2General Station of Animal Husbandry of Shandong Province, Jinan 250100, China; 13969155291@163.com (S.Z.); 18866617369@163.com (M.Z.); zhaixiangwei83@126.com (X.Z.); 3Protection of Animal Genetic Resources and Biological Breeding Engineering Research Center of Shandong Province, Jinan 250300, China

**Keywords:** Dapulian, Landrace, transcriptome, immune response, miRNA

## Abstract

Peripheral blood mononuclear cells (PBMCs) are crucial for the host’s immune system. This study analyzed the PBMCs transcriptome data from Dapulian (Chinese local breed) and Landrace (Commercial breed) pigs after stimulation with polyinosinic-polycytidylic acid. Differentially expressed genes were identified through comparative analysis, followed by gene ontology (GO) and Kyoto encyclopedia of genes and genomes (KEGG) enrichment analyses; pathways such as the MHC protein complex, the IL-17 signaling pathway and Toll-like receptor signaling pathway were enriched, alongside key immune response-related genes in Dapulian pig, such as *CXCL14*, *IL1R1*, *IL33*, and *SLA-DMA*. Furthermore, an miRNA–mRNA regulatory network was constructed to elucidate the immune response in Dapulian pig. These findings highlight the transcriptomic differences between pig breeds and offer insights into the immune response regulation of antiviral responses in indigenous and modern commercial pigs.

## 1. Introduction

Pigs are one of the most prevalent domestic animals around the world. Domestic pig (*Sus scrofa domesticus*) was domesticated by humans from wild boar in Europe and Asia around 10,000 years ago [1]. Over the past few decades, with the development of intensive breeding technology, large-scale group breeding has become a model for the development of modern animal husbandry, which has resulted in a series of immune problems [2,3]. In the pig industry, diseases caused by viral or bacterial pathogens exert a limitation on growth efficiency, and infectious diseases usually cause huge economic losses, and seriously restrict the development of the industry, such as African swine fever (ASF) [4,5]. As is well known, the indigenous pig breeds in China have better disease resistance than commercial pig breeds [6]. Thus, a deeper understanding of the comprehensive regulatory network that controls disease resistance in pigs has become an urgent research topic in farm animal husbandry.

In pigs, disease resistance is usually composed of two aspects, the general and the specific anti-disease ability. The former refers to the pathogen resistance of pigs, that is, the collective comprehensive defense ability against various pathogens, and it mainly involves immune factors and immune cells in the body. The latter refers to the specific resistance of pigs to specific diseases or pathogens, and it depends on the body’s immune response. Peripheral blood mononuclear cells (PBMCs), belonging to the first defense line of the immune system, contain a wide range of immune cells, and play an integral part in the rapid response of the immune system during bacterial or viral infections [7,8]. Numerous studies have shown that PBMCs are suitable for modeling the disease pathogenesis and basis genetics underlying infectious or irritant diseases [9,10,11]. Studying the transcriptional differences in PBMCs of pigs is beneficial to understanding the genetic basis of resistance to disease.

Polyinosinic-polycytidylic acid (poly I:C) is a synthetic viral double-stranded RNA (dsRNA), which can be recognized by Toll-like receptor 3, induces IFN production, and can be used to simulate viral infection in experiments [12,13]. Poly I:C can also activate IPS-1- and TRIF-dependent pathways, leading to the activation of NK cells, which are natural immune system effector cells and mainly responsible for immune surveillance, and play an important part in preventing viral infections [14]. Poly I:C has widespread applications in experimental research of viral infection. Christopher K. Tuggle utilized poly I:C and lipopolysaccharide (LPS) to stimulate porcine alveolar macrophages (PAMs), revealing that non-stimulated and stimulated macrophages differ in terms of gene expression influenced by changes in chromatin accessibility at active regulatory regions, primarily mediated by H3K27ac modifications [15]. Mei conducted miRNA sequencing with poly I:C-stimulated and porcine reproductive and respiratory syndrome virus (PRRSV)-infected PAMs, suggesting that, compared with the poly I:C group, PRRSV slightly altered the miRNA expression profile, and this study confirmed that PRRSV is able to silence the host innate immune response at the early phase of infection [16].

RNA-seq has evolved into a major tool for studying molecular biology since its inception. With the advancement of analysis protocols, it has become possible to study genetic changes between and within species [17,18]. In pig, it has revealed that the PBMC genes’ expression profiles were altered by Poly I:C in different pig breeds, and some genes that changed in the two breeds were involved in immune-related functions, which also play an important role in tissue development [18]. Here, we focus on the immune response of pigs from the transcriptomic aspect. The porcine immune response-related genes were obtained through differential gene expression comparative analysis; after the target genes’ identification and functional enrichment, the potential miRNA–mRNA regulatory network of immune response in Dapulian pig was constructed.

## 2. Materials and Methods

### 2.1. Data Collection

In the study, the transcriptome data of PBMCs from Dapulian and Landrace pigs were collected (control group of Dapulian (DC), control group of Landrace (LC), treatment group of Dapulian (DT), treatment group of Landrace (LT)) from public reports [18,19]. Six 5-week-old piglets were selected for blood sampling; all pigs were raised in the same facility and each group had three replicates. PBMCs treated with and without poly I:C infection were cultured for 24 h and collected for sequencing. All raw transcriptome data have been downloaded from the NCBI public database, and the transcriptome data accession numbers are as follows: mRNA—PRJNA301538 and miRNA—PRJNA308253.

### 2.2. Quality Control of Raw Data

To ensure the consistency of the processing results, the standard procedures of data preprocessing were adopted [20,21]. Firstly, raw RNA-seq data quality were assessed using FastQC (v0.11.9) [22]. The unqualified data, low-quality data and sequencing adapters were removed using the default parameters of FASTP (v0.22.0), and the obtained clean data were secondly checked by FastQC, then the high-quality data were adapted in subsequent data analyses [23].

### 2.3. Mapping to the Reference Genome

The porcine reference genome (GCF_000003025.6) was downloaded from the NCBI database and the high-quality data were aligned to the reference genome using STAR (v2.6.1b) software; the uniquely mapped reads were assembled into genes by utilizing the Ensembl Susscrofa annotation file (Sus_scrofa.Sscrofa11.1.111) [24]. The gene expression matrix was then constructed using featureCounts (v1.6.3) software [25]. For miRNA data, the Subjunc (v1.6.3) software was used for reads alignment, and the transcript expression matrix was generated and quantified using the featureCounts (v1.6.3) software [26]. In addition, the exonic regions were assigned for read counts, with the gene_id attribute serving as the unique identifier to aggregate exons belonging to the same genes. Fragments per kilobase of exon model per million mapped fragments (FPKM) were then calculated using a custom R script, which was applied for gene expression assessment. 

### 2.4. Principal Components Analysis (PCA) and Sample Correlation Analysis

The scale function of R was used for normalization, and the prcomp function computes the principal components of standardized gene count by constructing a correlation matrix. For numerical stability, a singular value decomposition (SVD) is employed rather than the Eigen decomposition approach, and ggplot2 (R package, v3.4.3) is used for visualization. The cor function in the R environment was used to calculate the correlation values between samples, and pheatmap (R package, v1.0.12) was used for visualization. 

### 2.5. Identification of Differentially Expressed Genes

To obtain differentially expressed gene and miRNA information, DESeq2 (R package, v1.38.3) was applied, a typical differential analysis tool that employs the shrinkage estimation of dispersion and fold changes to enhance the stability and interpretability of results. The analysis begins with a count matrix *K*, where each row *i* corresponds to a gene, each column *j* represents a sample, and each entry *K_ij_* denotes the number of sequencing reads unambiguously mapped to gene *i* in sample *j* [27]. In the results, the fold change of each gene was calculated, as well as the false discovery rate-corrected *p*-value, and the genes with |Log2(fold change)| > 1 and −log10(*p*-value) > 1.3 were regarded as differentially expressed genes (DEGs), while |Log2(fold change)| > 0 and −log10(*p*-value) > 1.3 are considered to be the threshold of differentially expressed miRNAs (DEMs); the results are displayed in volcano plots.

### 2.6. Gene Functional Enrichment

With gene symbol as the input, the functional annotation of the target gene set was performed using clusterProfiler (R package, v4.6.2) [28], and the terms were considered significant with a *p*-value < 0.05, plotted using ggplot2 (R package, v3.4.3) and enrichplot (R package, v1.18.4).

### 2.7. Network Analysis of Protein-Protein Interaction (PPI)

The PPI network of Landrace immune response-related genes in Appendix A was performed using the string (v12.0) online URL https://string-db.org/; the minimum required interaction score is 0.900, and the meaning of network edges represents confidence, with line thickness indicating the strength of data support.

### 2.8. Target Genes of miRNA Prediction

To analyze the targeting relationship between miRNAs and genes, the miRanda database was used for target gene predictions of miRNA. Firstly, the sequence alignment was performed based on sequence complementarity, followed by calculating the minimum free energy (MFE) score for each selected miRNA–mRNA pair, and the pairs with MFE scores exceeding the predefined threshold are finally selected as potential targets [29]. Subsequently, the g:Profiler tool (https://biit.cs.ut.ee/gprofiler/gost, accessed on 22 July 2024) was used to convert the gene names.

## 3. Results

### 3.1. The Different Transcriptome Modes of PBMCs with and Without Poly I:C Treatment in Pig

To explore the molecular mechanisms of immune response in native and modern commercial pigs, we collected the public transcriptome data of PBMCs from Dapulian and Landrace pigs, divided into four groups according to the poly I:C infection (treatment group of Dapulian (DT), treatment of Landrace (LT)) and not (control of Dapulian (DC), control of Landrace (LC)), and the underlying transcriptome dynamics were investigated (Figure 1A). Firstly, we performed principal component analysis and correlation analysis with all the investigated samples, and found that the PBMC samples from different pig species could be clearly separated (Figure 1B), while the control group was different from that of the poly I:C-infected group (Figure 1C and Appendix A).

### 3.2. The DEGs Identification of PBMCs from Different Groups

To obtain insights into the effects of poly I:C on the transcriptome characteristics of PBMCs in pig, a differential analysis was performed. A total of 757 DEGs were generated in Dapulian pigs, including 607 upregulated and 150 downregulated genes (Figure 2A and Appendix A). A total of 128 DEGs were obtained in Landrace; 55 genes were increased and 73 declined (Figure 2B and Appendix A).

Functional enrichment analysis was then performed using gene ontology (GO) and Kyoto encyclopedia of genes and genomes (KEGG). For the DEGs in Dapulian pigs, the GO enrichment analysis showed they were enriched in immunological functions, such as the processes of inflammatory response, cytokine activity, the regulation of immune system processes, the regulation of lymphocyte activation, and MHC protein complex (Figure 2C and Appendix A), and in the result of KEGG enrichment analysis, these DEGs were also found to be related with immunological pathways, including viral protein interactions with cytokine and cytokine receptor, the IL-17 signaling pathway, the intestinal immune network for IgA production, the chemokine signaling pathway, and the Toll-like receptor signaling pathway (Figure 2E and Appendix A). Regarding the DEGs in Landrace, they were involved in immune-related functions, such as the MHC protein complex, adaptive immune response, inflammatory response, innate immune response, and the positive regulation of immune system processes (Figure 2D and Appendix A), and KEGG enrichment analysis showed that they were also enriched in immune system-related pathways, such as Viral protein interaction with the cytokine and cytokine receptor, cytokine–cytokine receptor interaction, the intestinal immune network for IgA production, the chemokine signaling pathway, and antigen processing and presentation (Figure 2F and Appendix A).

### 3.3. Key Regulatory Genes of Immune Response in Pigs

In order to identify the key regulatory genes of the immune response when pigs are infected by poly I:C, we found a total of 40 DEGs that were differentially expressed both in Dapulian and Landrace pigs (Figure 3A and Appendix A), such as *VCAM1*, *CLDN4* and *C1S*.

Then, GO enrichment analysis found that these genes were also enriched in immune response-related pathways, such as adaptive immune response, antigen processing and presentation, antigen processing and presentation of the peptide or polysaccharide antigen via MHC class II, antigen processing and the presentation of exogenous antigen, and the immune effector process (Figure 3B and Appendix A). In addition, these DEGs were also subjected to KEGG enrichment analysis, and it revealed that *VCAM1* and *CLDN4* were involved in the pathway of leukocyte transendothelial migration, *C1S* and *C1R* were related with complement and coagulation cascades, *TFRC* plays a role in phagosome process, and *SLA-DMB*, *SLA-DRB1* and *SLA-DRA* were enriched in terms of the intestinal immune network for IgA production, antigen processing and presentation, Th1 and Th2 cell differentiation, and Th17 cell differentiation, etc. (Figure 3C and Appendix A).

### 3.4. Functional Enrichment of Genes Specifically Expressed in Dapulian and Landrace

To explore the functions of DEGs specifically expressed in Dapulian and Landrace, we performed functional enrichment analyses on 717 and 88 DEGs unique to each species, respectively (Appendix A).

Through the GO enrichment analysis of 717 DEGs specifically in Dapulian, we found that the inflammatory response, cytokine activity, cytokine receptor binding, regulation of leukocyte activation, and regulation of immune system process were enriched (Figure 4A). KEGG analysis enriched cytokine–cytokine receptor interaction, viral protein interaction with cytokine and cytokine receptor, the IL-17 signaling pathway, inflammatory bowel disease, and Th17 cell differentiation pathways (Figure 4B).

Similarly, the GO enrichment analysis of 88 DEGs specifically in Landrace revealed chemokine activity, inflammatory response, chemokine receptor binding, cytokine activity, and immune system development (Figure 4C). KEGG analysis enriched phagosome, viral protein interaction with cytokine and cytokine receptors, cytokine–cytokine receptor interaction, the chemokine signaling pathway, and the IL-17 signaling pathways (Figure 4D).

### 3.5. Differential Expression Analysis of miRNA in Different Groups

In addition, to characterize the changes of miRNA in control and treatment groups, a differential expression analysis of miRNA was conducted between DT and DC, as well as LT and LC groups (Appendix A). A total of 10 DEMs in Dapulian were obtained (Figure 5A)—8 DEMs (such as miR855-5p, miR1-3p and miR208b) were upregulated and 2 were downregulated (i.e., miR486 and miR181a). In Landrace, only miR1249 was increased and miR192 was decreased (Figure 5B).

To identify miRNA target genes and functions, we used the miRanda database to predict the target genes of DEMs (Figure 5C and Appendix A). The GO enrichment showed that the target genes regulated by DEMs in Dapulian were enriched in terms of interleukin-6 production, cytokine production, viral RNA genome replication and immune system development (Figure 5D and Appendix A). In addition, the target genes regulated by DEMs in Landrace were enriched in the process of regulation of cytokine production, T cell costimulation, the regulation of interleukin-6 production, the positive regulation of the toll-like receptor signaling pathway and the regulation of chemokine production pathways (Figure 5E and Appendix A).

### 3.6. Construction of Potential Regulatory Network of miRNA-mRNA Regard to Immune Response in Dapulian Pig

Given the interaction of genes and miRNA, the potential regulatory network of miRNA and genes was constructed in relation to the immune responses of pigs, which were based on the DEGs and DEMs in Dapulian. the expression levels of DEGs and DEMs were opposite to one another, because they are theoretically negatively regulated. We derived 73 miRNA–gene pairs (Figure 6A and Appendix A).

Firstly, regarding these 73 genes, we found they were enriched in lymphocyte apoptotic process, leukocyte apoptotic process, positive regulation of inflammatory response, cytokine activity and MHC protein complex (Figure 6B and Appendix A). In addition, based on the regulatory effects of miRNA on predicted targeting genes, we constructed the potential regulatory network of pairing miRNA–mRNA for the immune response process in Dapulian pigs. In the results, it was shown that ssc-miR133a-3p may regulate the *SLA-DMA* gene, and they were enriched in the Influenza A pathway. *CXCL2*, *FAS*, *IDO2*, *IL1R1*, *IL20*, *IL33*, *OSM* and *PTGS2* genes may be regulated by ssc-miR181a, and are enriched in the NF-kappa B signaling pathway, cytokine-cytokine receptor interaction, African trypanosomiasis, viral protein interaction with cytokine and cytokine receptor, and the Influenza A pathway. Furthermore, we have discovered the targeted interactions between ssc-miR208b and *CXCL2*, ssc-miR486 and *IL33* and *NFKB2*, and ssc-miR885-5p and *VCAM1*. These genes are implicated in pathways that play a role in immune modulation (Figure 6C and Appendix A).

## 4. Discussion

The environmental challenges brought about by the large-scale farming technology make disease resistance more and more important in pig production practice. The Dapulian pig is a local Chinese breed and has stronger disease resistance than commercial pigs [18]. Indeed, during the period of PRRSV infection, Duroc × Landrace × Yorkshire (DLY) pigs show a number of clinical features that are typical of the disease, while Dapulian pigs only show mild disease symptoms [30]. Here, the difference in immune response was investigated in Dapulian and Landrace pig based on the transcriptomic data of PBMCs from normal and Poly I:C-treated groups, and the miRNA–gene regulatory network potentially associated with immune response in Dapulian pig was constructed and assessed.

Firstly, the immune response of viral infection in pigs was investigated from the aspect of PBMCs transcriptomic data. Through gene differential expression analysis, 757 DEGs in Dapulian (such as *TIMD4*, *RNF128* and *IL1R2*) were obtained, and 128 DEGs in Landrace (such as *TIMD4*, *CPM* and *CXCL13*) were related to immune response. Among these genes, *XCL1* and *CXCL13* are chemokines that inhibit the apoptosis of immune cells during viral infection and play an important part in inflammatory response and immune regulation [31,32,33]. *IL1R2* is an anti-inflammatory antagonist that can prevent immune disorders and systemic inflammation caused by IL1. It is naturally present on neutrophils, B cells, monocytes and macrophages, and is associated with various inflammatory diseases, such as arthritis and ulcerative colitis [34]. The *TIMD4* gene is differentially expressed both in Dapulian and Landrace pigs. As a member of the cell immunoglobulin domain and mucin domain (TIM) gene family, it is a phosphatidylserine receptor, and plays an integral role in the immune response [35]. Gao et al. showed that *TIMD4* can be inhibited by macrophages in vitro and prevent the production of cytokines through the NF-κB signaling pathway [36]. In addition, RNF128, also known as GRAIL, is an E3 ubiquitin ligase, which is rapidly activated to be expressed in anergic T cells when infections happen [37]. Gao et al. showed that virus infection induces RNF128 expression, which interacts with TBK1 via E3 ubiquitin ligase, enhancing K63-linked ubiquitination in response to RNA or DNA viruses. A deficiency in RNF128 impairs the phosphorylation of IRF3, IFN-β production, K63-linked ubiquitination, and TBK1 activation [38]. Tian et al. also found that RNF128 can inhibit the inflammatory response as a negative regulator of IL-3/STAT5 signaling. In addition, RNF128 facilitates the lysosomal degradation of IL-3Rα by catalyzing its K27-linked polyubiquitination in a ligase activity-dependent fashion. This finding also confirmed that RNF128 could inhibit macrophage activation and chemotaxis, thereby reducing excessive inflammatory response [39].

Specially, from the difference in the results regarding immune response genes, we may obtain some clues regarding the higher disease resistance of Dapulian pigs compared to western commercial pig breeds. Studies have shown that the superior immune function of Dapulian pigs compared to commercial pigs may be associated with the expressions of genes such as *CXCL8*; upon stimulation with oligodeoxynucleotides containing unmethylated CpG motifs (CpG ODN), the mRNA levels of the cytokine IL-8 (CXCL8) in the PBMCs of Dapulian pigs were significantly higher than those observed in Landrace pigs [40]. Hui Wang et al. also found that the expression levels of *ADAM8* and *CD59* genes in whole blood were significantly lower in Landrace compared to Dapulian piglets at the neonatal stage [41]. In our results, the *CXCL8*, *ADAM8*, and *CD59* genes were upregulated in DT vs. DC groups, but showed no significant differences in LT vs. LC groups. This further supports the hypothesis that these three genes may be associated with the higher immunity observed in Dapulian pigs. In addition, our results reveal that *ISG20* and *IRF1* were upregulated in the DT vs. DC group, but their expressions showed no significant differences in the LT vs. LC group. The *ISG20* gene, known as an interferon-stimulated gene (*ISG*), exhibits increased expression levels consistent with the findings of Bang Liu et al. Furthermore, *IRF1* is a member of the interferon regulatory factors (IRFs) family; PRRSV can suppress the expression of antiviral genes by disrupting *IRF1* gene transcription [42,43]. Therefore, the differential expressions of *ISG20* and *IRF1* genes may be one of the reasons for the varying immune responses observed between Dapulian and Landrace pigs.

The differential expressions suggest some miRNAs may be involved in the immune responses in pig, such as ssc-miR486, ssc-miR-181a, ssc-miR-885-5p, ssc-miR208b and ssc-miR192. As reported, the differential expressions of miRNA abundance in lung tissues of PCV2 (porcine circovirus type 2)-infected and PCV2-uninfected LW (Laiwu, a Chinese indigenous pig) and YL (Yorkshire × Landrace) pigs were investigated, and the qRT-PCR validation revealed that there were significant differences in the expressions of ssc-miR486 and ssc-miR-192 before and after infection [44]. Another study found that increasing miR-181a expression in mature T cells enhanced their sensitivity to peptide antigens, whereas inhibiting expression in immature T cells reduced sensitivity [45]. Also, ssc-miR-181a was differentially expressed in the duodenum of E. coli F18-sensitive and -resistant weaned piglets, which was verified by qRT-PCR [46]. Moreover, ssc-miR-885-5p was found to play an immune role in PDCoV (porcine del-tacoronavirus) infection in pigs through competitively binding mRNAs with lncRNA [47]; miR-208b is capable of inhibiting type I IFN signaling in HCV (Hepatitis C vi-rus)-infected hepatocytes, and the inhibition of miR-208b with miRNA inhibitors during HCV infection increased the expression of IFNAR1 (interferon alpha and beta receptor subunit 1). In our results, ssc-miR208b expression was significantly different in the DC and DT groups, which may influence the immune reaction in Dapulian pigs [48].

In addition, a potential regulatory network of miRNA-mRNA in the immune response process was constructed in Dapulian, which included five miRNAs and 12 genes. Among them, *CXCL2* and *CXCL14* are chemokines, whose expression levels change when they are infected with a virus [49]. Interleukin 20 (*IL-20*) and interleukin 33 (*IL-33*) are both cytokines. *IL-33* is a tissue-derived nuclear cytokine that belongs to the *IL-1* family of cytokines, which acts as a trigger for alarm signals during cell damage or tissue damage, and sends a warning to immune cells expressing the ST2 receptor (*IL-1RL1*) [50]. *VCAM1* is a cell adhesion molecule, and IL1R1 is an IL-1 receptor; they play an essential role in inflammation and immunity [51,52]. The *MHC* gene is closely related to the immune system and called swine leukocyte antigen (*SLA*) in pigs [53,54]. A study conducted on two pig breeds found that the *SLA* gene has a certain connection with porcine reproductive and respiratory syndrome, porcine pseudorabies and mycoplasma pneumonia [55]. It is suggested that these proteins are persistently expressed on the surfaces of almost every nucleated cell, serving to present peptides to CD8+ cytotoxic T cells and to interface with natural killer (NK) cells [56]. Some studies have also found that SLA haplotypes are not only related to pre-weaning mortality caused by diarrhea, but also differ in relation to parasite resistance [57,58].

Furthermore, there were certain defects and limitations in the study. In one aspect, the transcriptome data of PBMCs were derived from six 5-week-old piglets of Landrace (two male and one female) and Dapulian (one male and two female) breeds; each group included three repeats and the number of experiment animals was relatively limited. One repeat from the DT group was eliminated because of sample discretion, thus the results were weak from the point of view of statistical significance. Besides this, the results of this study were obtained via data analysis, and lack experimental verification. Further experiments are still required to confirm and strengthen the conclusions of this study.

## 5. Conclusions

In this study, based on a comparative analysis of transcriptome data in Dupulian pigs with and without poly I:C stimulation, the candidate genes associated with disease resistance, such as *TIMD4*, *RNF128*, and *IL1R2*, were highlighted, and the miRNA–gene regulatory network related to the immune response of Dapulian pigs was built. This study not only found changes in immune-related genes in Dapulian and Landrace pigs after virus infection, but also explored the specific changes in immune-related genes in Daplian, providing a new scientific basis for the future breeding of pig disease resistance.

## Figures and Tables

**Figure 1 animals-14-03546-f001:**
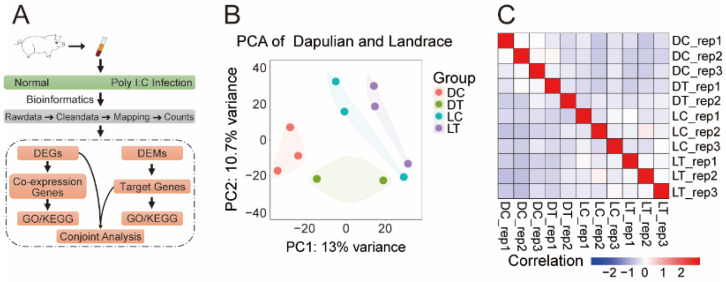
Transcriptome analysis of PBMCs from Dapulian and Landrace pigs in normal pigs and with poly I:C infection. (**A**) The technical route of data analysis in this study; (**B**) PCA of transcriptome data in four groups of Dapulian and Landrace; (**C**) correlation analysis of the PBMCs samples between Dapulian and Landrace from normal and poly I:C-infected groups.

**Figure 2 animals-14-03546-f002:**
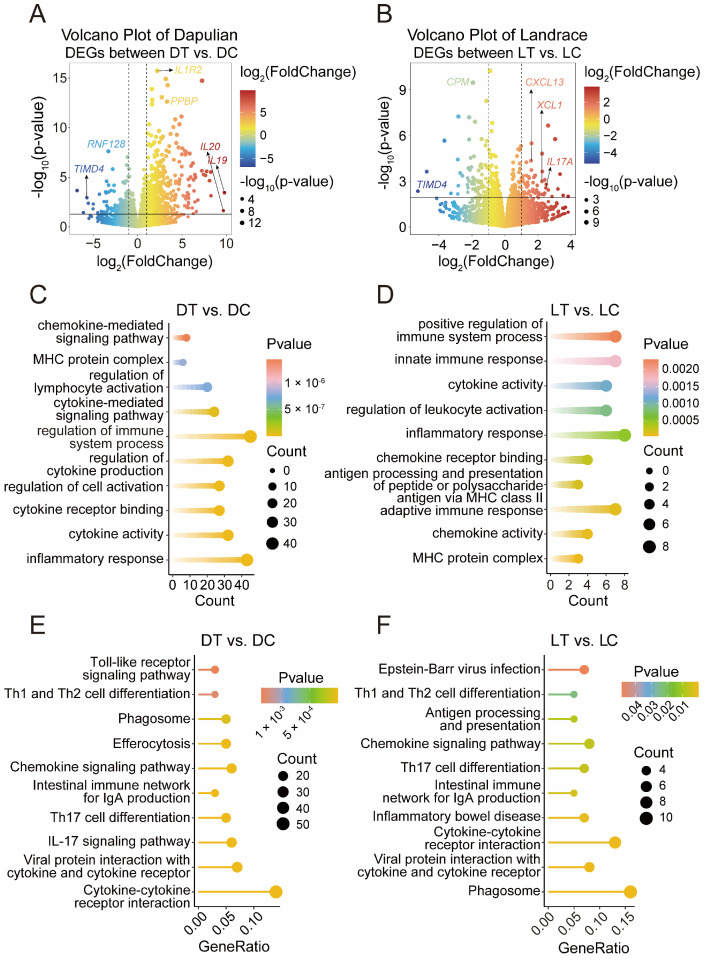
DEGs analysis of PBMCs from control and treatment groups. (**A**) Volcano plot of DEGs between DT and DC, where red represents upregulated genes, blue represents downregulated genes, and marker genes are related to immune function. (**B**) Volcano plot of DEGs between LT and LC, where red represents upregulated genes, blue represents downregulated genes, and marker genes are related to immune function. (**C**) GO enrichment terms of DEGs in Dapulian. (**D**) GO enrichment pathways of DEGs in Landrace. (**E**) KEGG enrichment terms of DEGs in Dapulian. (**F**) KEGG enrichment pathways of DEGs in Landrace.

**Figure 3 animals-14-03546-f003:**
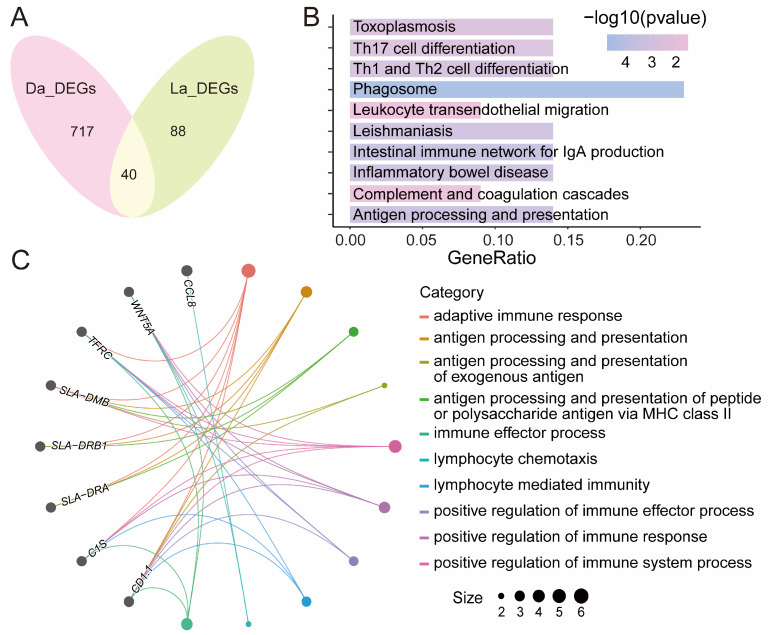
Identification of key regulatory genes of immune response in pigs. (**A**) Venn diagram of overlapping DEGs in Dapulian and Landrace. (**B**) GO enrichment analysis of co-expressed DEGs between Dapulian and Landrace. (**C**) KEGG enrichment of co-expressed DEGs between Dapulian and Landrace.

**Figure 4 animals-14-03546-f004:**
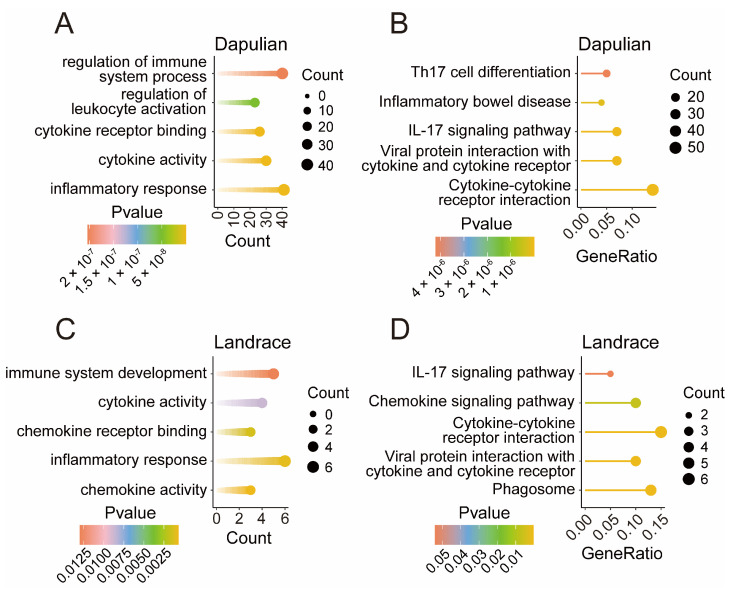
Functional enrichment of genes specifically expressed in Dapulian and Landrace. (**A**) GO enrichment analysis of specific DEGs in Dapulian. (**B**) KEGG enrichment analysis of specific DEGs in Dapulian. (**C**) GO enrichment analysis of specific DEGs in Landrace. (**D**) KEGG enrichment analysis of specific DEGs in Landrace.

**Figure 5 animals-14-03546-f005:**
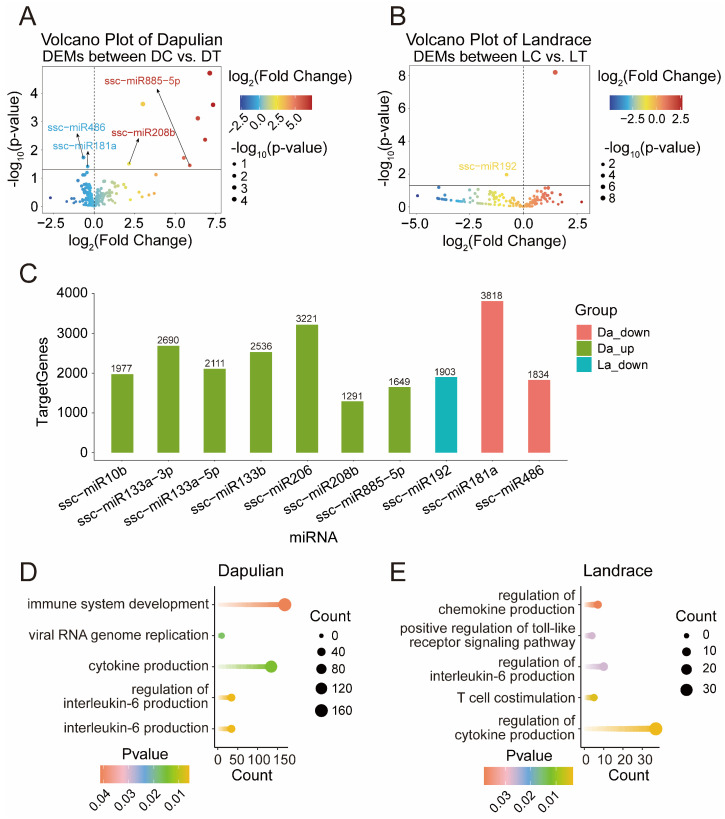
Analysis of DEMs in the PBMCs control group and treatment group. (**A**) Volcano plot of DEMs between DT and DC; the labeled miRNAs are the miRNAs included in the potential regulatory network of miRNA–mRNA below. (**B**) Volcano plot of DEMs between LT and LC. (**C**) Number of predicted target genes of DEMs in Dapulian and Landrace; the horizontal axis is the miRNA and the vertical axis is the number of target genes predicted by DEMs. (**D**) GO enrichment analysis of DEM target genes in Dapulian. (**E**) GO enrichment analysis of DEM target genes in Landrace.

**Figure 6 animals-14-03546-f006:**
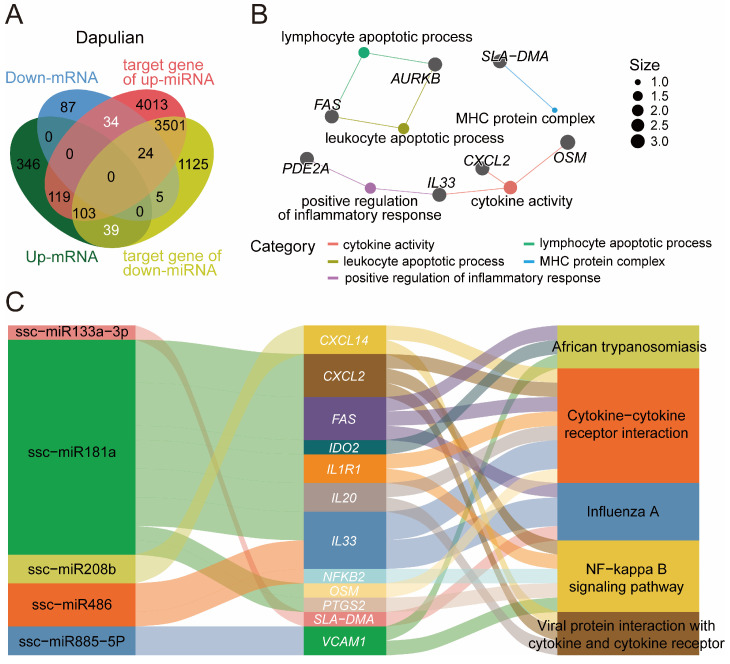
Construction of a potential regulatory network of miRNA–mRNA in the immune response of Dapulian pigs. (**A**) Venn diagram of DEGs and DEMs in Dapulian. (**B**) GO enrichment analysis of downregulated DEGs, upregulated DEM target genes, upregulated DEGs, and downregulate DEM target genes in Dapulian. (**C**) Targeting relationship between miRNA and genes in Dapulian.

## Data Availability

Not applicable.

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
