# Peer review of "Whole-Transcriptome Analysis Reveals the Regulatory Network of Immune Response in Dapulian Pig"

_animals, 2024, doi:10.3390/ani14233546_

Round 1
Reviewer 1 Report
Comments and Suggestions for Authors
To be honest, I have read this manuscript twice with the appreciation of scientific merit and enlighten to the future work in this field.
I have only comment on the information of pigs that were used in this study. The author should provide data, for example, age and body weight, this will be very useful for the discussion part if the pigs are entered the puberty or mature stage.
Author Response
Comments 1: To be honest, I have read this manuscript twice with the appreciation of scientific merit and enlighten to the future work in this field. I have only comment on the information of pigs that were used in this study. The author should provide data, for example, age and body weight, this will be very useful for the discussion part if the pigs are entered the puberty or mature stage.
Response 1: We appreciate for your comments and it is very important. In our study, the transcriptome data of PBMCs with and without poly I:C infection from Dapulian and Landrace pig were collected. Six 5-wk-old piglets were selected for blood sample, all pigs were raised in the same facility and each group had three replicates. All raw transcriptome data were downloaded from NCBI public database (PRJNA301538, PRJNA308253), and we have added the information of pigs in Data collection part of Materials and Methods.
Reviewer 2 Report
Comments and Suggestions for Authors
This article "Whole transcriptome analysis reveals the regulatory network of 2 immune response in Dapulian pig" is very novel since there aren´t conclusive studies related to disease resistance with genetic analysis. Transcriptome evaluation is essential to predict disease resistance between commercial and Dapulian pigs.
Material and methods. It is not clear how the groups were formed. What is the number of pigs used in each group? Commercial pigs and Dapulian pigs?
How to know if the analyzes are statistically representative according to the peripheral blood results of one or several pigs.
Did they do any animal purity studies?
If animals were not pure breeds it is possible that the results will not be associated specifically with Dapulian and Landrace?.
Author Response
Comments : This article "Whole transcriptome analysis reveals the regulatory network of 2 immune response in Dapulian pig" is very novel since there aren´t conclusive studies related to disease resistance with genetic analysis. Transcriptome evaluation is essential to predict disease resistance between commercial and Dapulian pigs. Material and methods. It is not clear how the groups were formed. What is the number of pigs used in each group?Commercial pigs and Dapulian pigs? How to know if the analyzes are statistically representative according to the peripheral blood results of one or several pigs. Did they do any animal purity studies? If animals were not pure breeds it is possible that the results will not be associated specifically with Dapulian and Landrace?
Response : Thank you very much. In the study, through comparative analysis of PBMCs with and without poly I:C infection, we aim to find the difference of immune response between commercial and Dapulian pigs, and a miRNA-mRNA regulatory network of immune response was constructed in pig, which would provide novel insight of candidate genes for further disease resistance in pig.
Question 1: Material and methods. It is not clear how the groups were formed. What is the number of pigs used in each group? Commercial pigs and Dapulian pigs?
Reply 1: Thanks for your comment. In our study, four subgroups of transcriptome data were analyzed, including normal PBMCs from Landrace pigs (LC), PBMCs from Landrace after poly I:C infection (LT), normal PBMCs from Dapulian pigs (DC), and PBMCs from Dapulian after poly I:C infection (DT), with three replicates from each group. However, because one Dapulian pig after poly I:C infection had less than ideal replication, so it was excluded from the subsequent analysis to improve the data reliability.
Question 2: How to know if the analyzes are statistically representative according to the peripheral blood results of one or several pigs.
Reply 2: Thanks for your professional review, the comment is important. In the study, the transcriptome data were collected from the public reports [1,2], as stated, three Landrace (two male and one female) and three Dapulian individuals (one male and two female) were applied. Given the cost of animal testing, three biological repeats in each group were designed, which may weaken the statistical significance of the results. Another, the results were mainly generated based on combined analysis of gene and microRNA data, which has been reported separately [1,2]. Here, we have added the content of the study limitation in Discussion, and the further experiment would be required to verify the conclusion.
Ref:
1. Wang, J.; Wang, Y.; Wang, H.; Wang, H.; Liu, J.-F.; Wu, Y.; Guo, J. Transcriptomic analysis identifies candidate genes and gene sets controlling the response of porcine peripheral blood mononuclear cells to poly I: C stimulation. G3: Genes, Genomes, Genetics 2016, 6, 1267-1275.
2. Wang, J.; Wang, Y.; Wang, H.; Guo, J.; Wang, H.; Wu, Y.; Liu, J. MicroRNA Transcriptome of Poly I:C-Stimulated Peripheral Blood Mononuclear Cells Reveals Evidence for MicroRNAs in Regulating Host Response to RNA Viruses in Pigs. Int J Mol Sci 2016, 17.
Question 3: Did they do any animal purity studies? If animals were not pure breeds it is possible that the results will not be associated specifically with Dapulian and Landrace?
Reply 3: Thanks for your professional review. Actually, we collected the data from the public reports (refs as showed above), which conducted by two professional institutes, Institute of Animal Science and Veterinary Medicine of Shandong Academy of Agricultural Sciences and College of Animal Science and Technology of China Agricultural University, as they stated in Materials and Methods, “Six 5-wk-old piglets, selected from one modern commercial breed (Landrace), and one Chinese indigenous breed (Dapulian), were used in the study. All the piglets were raised in the same facility, and had not received any vaccinations except one for classical swine fever (CSF) 21 d after birth.” Thus, although we are not sure whether animal purity studies have been done, we tend to believe the animals used in the studies are pure breeds.
Reviewer 3 Report
Comments and Suggestions for Authors
In this study, Wang et al. investigated the immune response mechanisms between Dapulian pigs, a Chinese local breed, and Landrace pigs, a commercial breed, following stimulation with polyinosinic-polycytidylic acid from a transcriptomic perspective. The authors identified candidate genes associated with disease resistance, including TIMD4, RNF128, and IL1R2, and constructed a miRNA-gene regulatory network related to the immune response in Dapulian pigs. This research provides novel insights into the regulation of antiviral immune responses in both indigenous and modern commercial pig breeds. Overall, the logic of this study is sound and the data analysis is credible. However, some issues need to be addressed before the paper is published.
1. The background information of the data needs to be detailed, such as the age of the pigs, etc.
2. I noticed that in Figure 1C, there are only two repeats for DT, which needs to be explained and discussed.
3. In Figures 2A and B, it is necessary to label interferon or inflammation-related genes. If the changes are not significant, a discussion is required.
4. The role of the differentially expressed miRNAs identified in Figure 5 in regulating immune responses, particularly in the context of antiviral immunity, needs to be discussed.
5. Line 80, change PBMC to PBMCs
6. Please pay attention to the format of the references and check if they comply with the journal's requirements. I have noticed that some formats are inconsistent.
Author Response
Comments: In this study, Wang et al. investigated the immune response mechanisms between Dapulian pigs, a Chinese local breed, and Landrace pigs, a commercial breed, following stimulation with polyinosinic-polycytidylic acid from a transcriptomic perspective. The authors identified candidate genes associated with disease resistance, including TIMD4, RNF128, and IL1R2, and constructed a miRNA-gene regulatory network related to the immune response in Dapulian pigs. This research provides novel insights into the regulation of antiviral immune responses in both indigenous and modern commercial pig breeds. Overall, the logic of this study is sound and the data analysis is credible. However, some issues need to be addressed before the paper is published.
- The background information of the data needs to be detailed, such as the age of the pigs, etc.
- I noticed that in Figure 1C, there are only two repeats for DT, which needs to be explained and discussed.
- In Figures 2A and B, it is necessary to label interferon or inflammation-related genes. If the changes are not significant, a discussion is required.
- The role of the differentially expressed miRNAs identified in Figure 5 in regulating immune responses, particularly in the context of antiviral immunity, needs to be discussed.
- Line 80, change PBMC to PBMCs
- Please pay attention to the format of the references and check if they comply with the journal's requirements. I have noticed that some formats are inconsistent.
Response: Thank you very much. In the present study, the transcriptome data of PBMCs from Dapulian and Landrace pigs were re-analyzed, the candidate genes associated with disease resistance were revealed, and a miRNA-gene regulatory network related to the immune response in pig was conducted, which may provide the novel insights regard to the difference of immune response between indigenous and modern commercial pig breeds. And the results are still required the further experimental verification.
Question 1: The background information of the data needs to be detailed, such as the age of the pigs, etc.
Reply 1: Thank you very much for the comment. In the study, the data were collected from public reports, as they descripted, the pigs were all 5-wk-old piglets, and 20 mL blood was collected from each piglet, PBMC were isolated following the manufacturer’s instructions and cultured at the cell concentration about 2 × 106/ml. During the culture, the PBMCs were treated with 20 mg/ml poly I:C in experiment group, and the control group was not, which were collected after 24 hours culture for sequencing. This content has also been added to the Data collection in the manuscript.
Question 2. I noticed that in Figure 1C, there are only two repeats for DT, which needs to be explained and discussed.
Reply 2: Thanks for your comment. For the question, in the data analysis process, we found that one sample from DT group is obviously deviates from other samples both in PCA and sample correlation analysis results, so in the process of data analysis, we eliminated it, but the later analysis we carried out strict quality control of the data, so as to improve the reliability of the results (lines 382 to 389).
“Furthermore, …, each group included three repeats and the number of experiment animal was is relatively limited. And one repeat from DT group was eliminated because of sample discrete, thus the results were weak from the statistical significance. Besides, the results of this study are basically obtained by data analysis, which is lack of the experimental verification. Comprehensively, it still needs to further experiments to confirm and strength the conclusions of this study.”
Question 3: In Figures 2A and B, it is necessary to label interferon or inflammation-related genes. If the changes are not significant, a discussion is required.
Reply 3: Thank you very much, your comment is great. In the figure 2A and 2B, we have marked some genes (such as TIMD4, RNF128, IL1R2, CPM and CXCL13) with significant changes related to immunity and discussed them in the Discussion (lines 303 to 327).
“Firstly, the immune response of viral infection in pig was investigated by using Poly I:C treatment from the aspect of PBMCs transcriptomic data. Through gene dif-ferential expression analysis, 757 DEGs in Dapulian (such as TIMD4, RNF128 and IL1R2) are obtained, and 128 DEGs in Landrace (such as TIMD4, CPM and CXCL13), which were related to immune response. Among these genes, PPBP (also termed CXCL7), XCL1 and CXCL13 are chemokines that inhibit the apoptosis of immune cells during viral infection and play an important part in inflammatory response and im-mune regulation [1-3]. IL1R2 is an anti-inflammatory antagonist that can prevent immune disorders and systemic inflammation caused by IL1. It is naturally present on neutrophils, B cells, monocytes and macrophages and is associated with various in-flammatory diseases, such as arthritis and ulcerative colitis [4]. TIMD4 gene is dif-ferentially expressed in Dapulian and Landrace pigs. As a member of T cell immuno-globulin domain and mucin domain (TIM) gene family, it is a phosphatidylserine re-ceptor and plays an integral role in the immune response [5]. And Gao et al. showed that TIMD4 can be inhibited by macrophages in vitro and prevent the production of cytokines through NF-κB signaling pathway [6]. RNF128, also known as GRAIL, is an E3 ubiquitin ligase, which is rapidly activated to express in anergic T cells when infec-tion happen [7]. Gao et al. showed that virus infection induces RNF128 expression, which interacts with TBK1 via E3 ubiquitin ligase, enhancing K63-linked ubiquitina-tion in response to RNA or DNA viruses. A deficiency in RNF128 impairs the phos-phorylation of IRF3, IFN-β production, K63-linked ubiquitination, and TBK1 activation [8]. Tian et al. also found that RNF128 can inhibit the inflammatory response as a negative regulator of IL-3/STAT5 signaling. In addition, RNF128 facilitates the lysosomal degradation of IL-3Rα by catalyzing its K27-linked polyubiquitination in a ligase activity-dependent fashion. This finding also confirmed that RNF128 could inhibit macrophage activation and chemotaxis, thereby reducing excessive inflammatory response [9].”
Ref:
1. Song, G.; Zhang, Y.; Gao, H.; Fu, Y.; Chen, Y.; Yin, Y.; Xu, K. Differences in Immune Characteristics and Related Gene Expression in Spleen among Ningxiang, Berkshire Breeds and Their Hybrid Pigs. Genes (Basel) 2024, 15.
2. Liu, G.; Wang, Y.; Jiang, S.; Sui, M.; Wang, C.; Kang, L.; Sun, Y.; Jiang, Y. Suppression of lymphocyte apoptosis in spleen by CXCL13 after porcine circovirus type 2 infection and regulatory mechanism of CXCL13 expression in pigs. Vet Res 2019, 50, 17.
3. Lei, Y.; Takahama, Y. XCL1 and XCR1 in the immune system. Microbes Infect 2012, 14, 262-267.
4. Peters, V.A.; Joesting, J.J.; Freund, G.G. IL-1 receptor 2 (IL-1R2) and its role in immune regulation. Brain Behav Immun 2013, 32, 1-8.
5. Yuan, J.; Tang, Z.L.; Yang, S.; Cao, J.Y.; Li, K. Molecular characteristics of the porcine TIMD4 gene and its association analysis. Biochem Genet 2012, 50, 538-548.
6. Xu, L.; Zhao, P.; Xu, Y.; Gao, L.; Wang, H.; Jia, X.; Ma, H.; Liang, X.; Ma, C.; Gao, L. Tim-4 protects mice against lipopolysaccharide-induced endotoxic shock by suppressing the NF-kappaB signaling pathway. Lab Invest 2016, 96, 1189-1197.
7. Anandasabapathy, N.; Ford, G.S.; Bloom, D.; Holness, C.; Paragas, V.; Seroogy, C.; Skrenta, H.; Hollenhorst, M.; Fathman, C.G.; Soares, L. GRAIL: an E3 ubiquitin ligase that inhibits cytokine gene transcription is expressed in anergic CD4+ T cells. Immunity 2003, 18, 535-547.
8. Song, G.; Liu, B.; Li, Z.; Wu, H.; Wang, P.; Zhao, K.; Jiang, G.; Zhang, L.; Gao, C. E3 ubiquitin ligase RNF128 promotes innate antiviral immunity through K63-linked ubiquitination of TBK1. Nat Immunol 2016, 17, 1342-1351.
9. Yu, J.; Li, J.; Shen, A.; Liu, Z.; He, T.S. E3 ubiquitin ligase RNF128 negatively regulates the IL-3/STAT5 signaling pathway by facilitating K27-linked polyubiquitination of IL-3Ralpha. Cell Commun Signal 2024, 22, 254.
Question 4: The role of the differentially expressed miRNAs identified in Figure 5 in regulating immune responses, particularly in the context of antiviral immunity, needs to be discussed.
Reply 4: Thanks for your comment. Regard to the question, the role of the differentially expressed miRNAs identified in Figure 5 in regulating the immune response has been added to the Discussion (lines 348 to 365), and it reads as follows:
“Based on the differential expression, it suggested some miRNAs may involve in the immune response in pig, such as ssc-miR486, ssc-miR-181a, ssc-miR-885-5p, ssc-miR208b and ssc-miR192. As reported, the differential expressions of miRNA abundance in lung tissues of PCV2 (porcine circovirus type 2) -infected and PCV2-uninfected LW (Laiwu, a Chinese indigenous pig) and YL (Yorkshire × Landrace) pigs was investigated, the qRT-PCR validation revealed that there were sig-nificant differences in the expression of ssc-miR486 and ssc-miR-192 before and after infection [1]. Another study found that increasing miR-181a expression in mature T cells enhanced their sensitivity to peptide antigens, whereas inhibiting expression in immature T cells reduced its sensitivity [2]. Also, ssc-miR-181a was differentially expressed in the duodenum of E. coli F18-sensitive and -resistant weaned piglets, which was verified by qRT-PCR [3]. Moreover, ssc-miR-885-5p was found to play an immune role in PDCoV (porcine deltacoronavirus) infection in pigs through competi-tively binding mRNAs with lncRNA [4], miR-208b is able of inhibiting type I IFN sig-nalling in HCV (Hepatitis C virus)-infected hepatocytes, and inhibition of miR-208b with miRNA inhibitors during HCV infection increased the expression of IFNAR1. And in our results, ssc-miR208b expression was significantly different in the DC and DT groups, which may involve in immune reaction in Dapulian pig [5].”
Ref:
1. Zhang, P.; Wang, L.; Li, Y.; Jiang, P.; Wang, Y.; Wang, P.; Kang, L.; Wang, Y.; Sun, Y.; Jiang, Y. Identification and characterization of microRNA in the lung tissue of pigs with different susceptibilities to PCV2 infection. Vet Res 2018, 49, 18.
2. Li, Q.J.; Chau, J.; Ebert, P.J.; Sylvester, G.; Min, H.; Liu, G.; Braich, R.; Manoharan, M.; Soutschek, J.; Skare, P.; et al. miR-181a is an intrinsic modulator of T cell sensitivity and selection. Cell 2007, 129, 147-161.
3. Ye, L.; Su, X.; Wu, Z.; Zheng, X.; Wang, J.; Zi, C.; Zhu, G.; Wu, S.; Bao, W. Analysis of differential miRNA expression in the duodenum of Escherichia coli F18-sensitive and -resistant weaned piglets. PLoS One 2012, 7, e43741.
4. Tang, X.; Lan, T.; Wu, R.; Zhou, Z.; Chen, Y.; Sun, Y.; Zheng, Y.; Ma, J. Analysis of long non-coding RNAs in neonatal piglets at different stages of porcine deltacoronavirus infection. BMC Vet Res 2019, 15, 111.
5. Jarret, A.; McFarland, A.P.; Horner, S.M.; Kell, A.; Schwerk, J.; Hong, M.; Badil, S.; Joslyn, R.C.; Baker, D.P.; Carrington, M.; et al. Hepatitis-C-virus-induced microRNAs dampen interferon-mediated antiviral signaling. Nat Med 2016, 22, 1475-1481.
Question 5: Line 80, change PBMC to PBMCs
Reply 5: Thank you very much. We have revised in the manuscript.
Question 6: Please pay attention to the format of the references and check if they comply with the journal's requirements. I have noticed that some formats are inconsistent.
Reply 6: We very appreciate for your comment. We carefully checked the format of the references, and the revision have been made. Thanks again.
Reviewer 4 Report
Comments and Suggestions for Authors
Dear authors,
This article applied a series of bioinformatic tools to analyze gene expression under the stimulation of poly I:C in 2 porcine breeds and generate a regulatory network of immune response in both porcine breeds. This work has been clearly presented.
The main question addressed in this article is to find differences in immune response upon viral infection between 2 breeds, given the introduction that there is a consensus that indigenous pigs in China are more resistant than modern commercial pigs in disease resistance. The authors did not answer or loop back to this question in the results, discussion, or conclusion. My first comment was to request for the closure of the main question in this article.
The topic is not original since there are other publications with similar approaches to discover pathways of immune response upon infections. However, their question to reveal the difference in the immune system of 2 breeds upon viral infection is reasonable and may potentially bring some insights for development of vaccines.
The authors applied bioinformatic tools to ask questions specifically between 2 certain breeds. Although the approach is not novel, it suits in its title if the question the authors raised had been answered and addressed in the article.
The improvement of this article should include more explanation of the results. All the outputs have to be based on the analysis setting and the rationale behind them and the authors did not have the rationale of choosing the programs for analysis. These authors did not put much explanation of how to enrich and consolidate the results of interests or if the results fit in any previous research outcomes.
Regarding the conclusions, as mentioned above, the main drawback of this article is no sufficient explanation of the results and rationale of how to get the results, and no closure of the main question raised in the beginning of the article. I did not find any inappropriate references.
There are a few comments for your considerations:
1. What is the take home message after building a network of immune response upon viral infection?
2. Given that viral infection triggers a bunch of pathways, which ones would be closely related to support the statement that Dapulian pigs are more disease resistant than modern commercial pigs? Please add your response to the discussion.
3. Figure 2D and 2E were mislabeled with porcine breeds.
Author Response
Comments: Dear authors,
This article applied a series of bioinformatic tools to analyze gene expression under the stimulation of poly I:C in 2 porcine breeds and generate a regulatory network of immune response in both porcine breeds. This work has been clearly presented.
The main question addressed in this article is to find differences in immune response upon viral infection between 2 breeds, given the introduction that there is a consensus that indigenous pigs in China are more resistant than modern commercial pigs in disease resistance. The authors did not answer or loop back to this question in the results, discussion, or conclusion. My first comment was to request for the closure of the main question in this article.
The topic is not original since there are other publications with similar approaches to discover pathways of immune response upon infections. However, their question to reveal the difference in the immune system of 2 breeds upon viral infection is reasonable and may potentially bring some insights for development of vaccines.
The authors applied bioinformatic tools to ask questions specifically between 2 certain breeds. Although the approach is not novel, it suits in its title if the question the authors raised had been answered and addressed in the article.
The improvement of this article should include more explanation of the results. All the outputs have to be based on the analysis setting and the rationale behind them and the authors did not have the rationale of choosing the programs for analysis. These authors did not put much explanation of how to enrich and consolidate the results of interests or if the results fit in any previous research outcomes.
Regarding the conclusions, as mentioned above, the main drawback of this article is no sufficient explanation of the results and rationale of how to get the results, and no closure of the main question raised in the beginning of the article. I did not find any inappropriate references.
There are a few comments for your considerations:
- What is the take home message after building a network of immune response upon viral infection?
- Given that viral infection triggers a bunch of pathways, which ones would be closely related to support the statement that Dapulian pigs are more disease resistant than modern commercial pigs? Please add your response to the discussion.
- Figure 2D and 2E were mislabeled with porcine breeds.
Reply to Reviewer 4:
This article applied a series of bioinformatic tools to analyze gene expression under the stimulation of poly I:C in 2 porcine breeds and generate a regulatory network of immune response in both porcine breeds. This work has been clearly presented.
The main question addressed in this article is to find differences in immune response upon viral infection between 2 breeds, given the introduction that there is a consensus that indigenous pigs in China are more resistant than modern commercial pigs in disease resistance. The authors did not answer or loop back to this question in the results, discussion, or conclusion. My first comment was to request for the closure of the main question in this article.
Reply: Thanks, your comment is extremely right. Through the reference and our results checking, the relevant content related the main question of comparation between indigenous pigs and modern commercial pigs has been supplemented in Discussion par t (lines 328 to 347).
“Specially, from the difference result of immune response genes, we may obtain some clues for the higher disease resistance of Dapulian pig compared to western commercial pig breeds. Studies have shown that the superior immune function of Dapulian pigs com-pared to commercial pigs may be associated with the expression of genes such as CXCL8, upon stimulation with oligodeoxynucleotides containing un-methylated CpG motifs (CpG ODN), the mRNA levels of the cytokine IL-8 (CXCL8) in the PBMCs of Dapulian pigs were significantly higher than those observed in Landrace pigs [40]. Hui Wang et al. also found that the expression levels of ADAM8 and CD59 genes in whole blood were significantly lower in Landrace compared to Dapulian pig-lets at the neonatal stage [1]. In our results, the CXCL8, ADAM8, and CD59 genes were upregulated in DT vs. DC groups, but showed no significant differences in LT vs. LC groups. This further supports the hypothesis that these three genes may be associ-ated with the higher immunity observed in Dapulian pig. In addition, our results re-vealed that ISG20 and IRF1 were upregulated in the DT vs. DC group, but their ex-pression showed no significant difference in the LT vs. LC group. The ISG20 gene, known as an interferon-stimulated gene (ISG), exhibits increased expression levels consistent with the findings of Bang Liu et al. Furthermore, IRF1 is a member of the in-terferon regulatory factors (IRFs) family, PRRSV can suppress the expression of anti-viral genes by disrupting IRF1 gene transcription [2, 3]. Therefore, the differential expression of ISG20 and IRF1 genes may be one of the reasons for the varying immune responses observed between Dapulian and Landrace pigs.”
Ref:
1. Hu, J.; Yang, D.; Chen, W.; Li, C.; Wang, Y.; Zeng, Y.; Wang, H. Whole Blood Transcriptome Sequencing Reveals Gene Expression Differences between Dapulian and Landrace Piglets. Biomed Res Int 2016, 2016, 7907980.
2. Bao, D.; Wang, R.; Qiao, S.; Wan, B.; Wang, Y.; Liu, M.; Shi, X.; Guo, J.; Zhang, G. Antibody-dependent enhancement of PRRSV infection down-modulates TNF-alpha and IFN-beta transcription in macrophages. Vet Immunol Immunopathol 2013, 156, 128-134.
3. Liang, W.; Meng, X.; Zhen, Y.; Zhang, Y.; Hu, X.; Zhang, Q.; Zhou, X.; Liu, B. Integration of Transcriptome and Proteome in Lymph Nodes Reveal the Different Immune Responses to PRRSV Between PRRSV-Resistant Tongcheng Pigs and PRRSV-Susceptible Large White Pigs. Front Genet 2022, 13, 800178.
The topic is not original since there are other publications with similar approaches to discover pathways of immune response upon infections. However, their question to reveal the difference in the immune system of 2 breeds upon viral infection is reasonable and may potentially bring some insights for development of vaccines.
The authors applied bioinformatic tools to ask questions specifically between 2 certain breeds. Although the approach is not novel, it suits in its title if the question the authors raised had been answered and addressed in the article.
The improvement of this article should include more explanation of the results. All the outputs have to be based on the analysis setting and the rationale behind them and the authors did not have the rationale of choosing the programs for analysis. These authors did not put much explanation of how to enrich and consolidate the results of interests or if the results fit in any previous research outcomes.
Regarding the conclusions, as mentioned above, the main drawback of this article is no sufficient explanation of the results and rationale of how to get the results, and no closure of the main question raised in the beginning of the article. I did not find any inappropriate references.
Reply: The comment is constructive and critical to our study. Indeed, we aim to bring some novel finding for disease resistance in pig from the regulation of immune response. As you suggested, more explanation of our results has been made in the Results and Discussion, as well as comparation of our result with the previous research outcomes, to consolidate our conclusion. In addition, the detail information regard to parameter settings and principles in data analysis process have been added in manuscript (lines 328 to 365).
“Specially, from the difference result of immune response genes, we may obtain some clues for the higher disease resistance of Dapulian pig compared to western commercial pig breeds. Studies have shown that the superior immune function of Dapulian pigs com-pared to commercial pigs may be associated with the expression of genes such as CXCL8, upon stimulation with oligodeoxynucleotides containing un-methylated CpG motifs (CpG ODN), the mRNA levels of the cytokine IL-8 (CXCL8) in the PBMCs of Dapulian pigs were significantly higher than those observed in Landrace pigs [40]. Hui Wang et al. also found that the expression levels of ADAM8 and CD59 genes in whole blood were significantly lower in Landrace compared to Dapulian pig-lets at the neonatal stage [1]. In our results, the CXCL8, ADAM8, and CD59 genes were upregulated in DT vs. DC groups, but showed no significant differences in LT vs. LC groups. This further supports the hypothesis that these three genes may be associ-ated with the higher immunity observed in Dapulian pig. In addition, our results re-vealed that ISG20 and IRF1 were upregulated in the DT vs. DC group, but their ex-pression showed no significant difference in the LT vs. LC group. The ISG20 gene, known as an interferon-stimulated gene (ISG), exhibits increased expression levels consistent with the findings of Bang Liu et al. Furthermore, IRF1 is a member of the in-terferon regulatory factors (IRFs) family, PRRSV can suppress the expression of anti-viral genes by disrupting IRF1 gene transcription [2, 3]. Therefore, the differential expression of ISG20 and IRF1 genes may be one of the reasons for the varying immune responses observed between Dapulian and Landrace pigs.
“Based on the differential expression, it suggested some miRNAs may involve in the immune response in pig, such as ssc-miR486, ssc-miR-181a, ssc-miR-885-5p, ssc-miR208b and ssc-miR192. As reported, the differential expressions of miRNA abundance in lung tissues of PCV2 (porcine circovirus type 2) -infected and PCV2-uninfected LW (Laiwu, a Chinese indigenous pig) and YL (Yorkshire × Landrace) pigs was investigated, the qRT-PCR validation revealed that there were sig-nificant differences in the expression of ssc-miR486 and ssc-miR-192 before and after infection [4]. Another study found that increasing miR-181a expression in mature T cells enhanced their sensitivity to peptide antigens, whereas inhibiting expression in immature T cells reduced its sensitivity [5]. Also, ssc-miR-181a was differentially expressed in the duodenum of E. coli F18-sensitive and -resistant weaned piglets, which was verified by qRT-PCR [6]. Moreover, ssc-miR-885-5p was found to play an immune role in PDCoV (porcine deltacoronavirus) infection in pigs through competi-tively binding mRNAs with lncRNA [7], miR-208b is able of inhibiting type I IFN sig-nalling in HCV (Hepatitis C virus)-infected hepatocytes, and inhibition of miR-208b with miRNA inhibitors during HCV infection increased the expression of IFNAR1. And in our results, ssc-miR208b expression was significantly different in the DC and DT groups, which may involve in immune reaction in Dapulian pig [8].”
Ref:
1. Hu, J.; Yang, D.; Chen, W.; Li, C.; Wang, Y.; Zeng, Y.; Wang, H. Whole Blood Transcriptome Sequencing Reveals Gene Expression Differences between Dapulian and Landrace Piglets. Biomed Res Int 2016, 2016, 7907980.
2. Bao, D.; Wang, R.; Qiao, S.; Wan, B.; Wang, Y.; Liu, M.; Shi, X.; Guo, J.; Zhang, G. Antibody-dependent enhancement of PRRSV infection down-modulates TNF-alpha and IFN-beta transcription in macrophages. Vet Immunol Immunopathol 2013, 156, 128-134.
3. Liang, W.; Meng, X.; Zhen, Y.; Zhang, Y.; Hu, X.; Zhang, Q.; Zhou, X.; Liu, B. Integration of Transcriptome and Proteome in Lymph Nodes Reveal the Different Immune Responses to PRRSV Between PRRSV-Resistant Tongcheng Pigs and PRRSV-Susceptible Large White Pigs. Front Genet 2022, 13, 800178.”
4. Zhang, P.; Wang, L.; Li, Y.; Jiang, P.; Wang, Y.; Wang, P.; Kang, L.; Wang, Y.; Sun, Y.; Jiang, Y. Identification and characterization of microRNA in the lung tissue of pigs with different susceptibilities to PCV2 infection. Vet Res 2018, 49, 18.
5. Li, Q.J.; Chau, J.; Ebert, P.J.; Sylvester, G.; Min, H.; Liu, G.; Braich, R.; Manoharan, M.; Soutschek, J.; Skare, P.; et al. miR-181a is an intrinsic modulator of T cell sensitivity and selection. Cell 2007, 129, 147-161.
6. Ye, L.; Su, X.; Wu, Z.; Zheng, X.; Wang, J.; Zi, C.; Zhu, G.; Wu, S.; Bao, W. Analysis of differential miRNA expression in the duodenum of Escherichia coli F18-sensitive and -resistant weaned piglets. PLoS One 2012, 7, e43741.
7. Tang, X.; Lan, T.; Wu, R.; Zhou, Z.; Chen, Y.; Sun, Y.; Zheng, Y.; Ma, J. Analysis of long non-coding RNAs in neonatal piglets at different stages of porcine deltacoronavirus infection. BMC Vet Res 2019, 15, 111.
8. Jarret, A.; McFarland, A.P.; Horner, S.M.; Kell, A.; Schwerk, J.; Hong, M.; Badil, S.; Joslyn, R.C.; Baker, D.P.; Carrington, M.; et al. Hepatitis-C-virus-induced microRNAs dampen interferon-mediated antiviral signaling. Nat Med 2016, 22, 1475-1481.
There are a few comments for your considerations:
Question 1: What is the take home message after building a network of immune response upon viral infection?
Reply 1: Thank you very much for your evaluation, your professional suggestion is helpful for our manuscript. As we introduced in Introduction, “the disease resistance is usually consisted of two aspects, the general and specific anti-disease ability in pig, the former refers to the pathogen resistance of pigs, that is the collective comprehensive defense ability against various pathogens, and it mainly involve in immune factors and immune cells in the body.” (lines 51 to 54) In the study, the immune response network was investigated after viral infection, which was stimulated with Ploy I:C in vitro, in our result, the expression of immune-related genes is differently altered between Dapulian (Chinese local breed) and Landrace (Commercial breed) pigs, and we have also identified some miRNAs that may influence immune regulation in pigs, and these findings provide a scientific basis for disease resistance in pig from the regulation of immune response.
Question 2: Given that viral infection triggers a bunch of pathways, which ones would be closely related to support the statement that Dapulian pigs are more disease resistant than modern commercial pigs? Please add your response to the discussion.
Reply 2: Thank you very much for your comment. For the question, the related content has been added to Discussion as follows (lines 328 to 347):
“Specially, from the difference result of immune response genes, we may obtain some clues for the higher disease resistance of Dapulian pig compared to western commercial pig breeds. Studies have shown that the superior immune function of Dapulian pigs com-pared to commercial pigs may be associated with the expression of genes such as CXCL8, upon stimulation with oligodeoxynucleotides containing un-methylated CpG motifs (CpG ODN), the mRNA levels of the cytokine IL-8 (CXCL8) in the PBMCs of Dapulian pigs were significantly higher than those observed in Landrace pigs [40]. Hui Wang et al. also found that the expression levels of ADAM8 and CD59 genes in whole blood were significantly lower in Landrace compared to Dapulian pig-lets at the neonatal stage [1]. In our results, the CXCL8, ADAM8, and CD59 genes were upregulated in DT vs. DC groups, but showed no significant differences in LT vs. LC groups. This further supports the hypothesis that these three genes may be associ-ated with the higher immunity observed in Dapulian pig. In addition, our results re-vealed that ISG20 and IRF1 were upregulated in the DT vs. DC group, but their ex-pression showed no significant difference in the LT vs. LC group. The ISG20 gene, known as an interferon-stimulated gene (ISG), exhibits increased expression levels consistent with the findings of Bang Liu et al. Furthermore, IRF1 is a member of the in-terferon regulatory factors (IRFs) family, PRRSV can suppress the expression of anti-viral genes by disrupting IRF1 gene transcription [2, 3]. Therefore, the differential expression of ISG20 and IRF1 genes may be one of the reasons for the varying immune responses observed between Dapulian and Landrace pigs.”
Ref:
1. Hu, J.; Yang, D.; Chen, W.; Li, C.; Wang, Y.; Zeng, Y.; Wang, H. Whole Blood Transcriptome Sequencing Reveals Gene Expression Differences between Dapulian and Landrace Piglets. Biomed Res Int 2016, 2016, 7907980.
2. Bao, D.; Wang, R.; Qiao, S.; Wan, B.; Wang, Y.; Liu, M.; Shi, X.; Guo, J.; Zhang, G. Antibody-dependent enhancement of PRRSV infection down-modulates TNF-alpha and IFN-beta transcription in macrophages. Vet Immunol Immunopathol 2013, 156, 128-134.
3. Liang, W.; Meng, X.; Zhen, Y.; Zhang, Y.; Hu, X.; Zhang, Q.; Zhou, X.; Liu, B. Integration of Transcriptome and Proteome in Lymph Nodes Reveal the Different Immune Responses to PRRSV Between PRRSV-Resistant Tongcheng Pigs and PRRSV-Susceptible Large White Pigs. Front Genet 2022, 13, 800178.
Question 3: Figure 2D and 2E were mislabeled with porcine breeds.
Reply 3: Thanks for your comment. The reminder is timely and important, we have a check in the whole manuscript, the relevant content has been corrected.
Round 2
Reviewer 3 Report
Comments and Suggestions for Authors
The manuscript has been revised in accordance with the reviewer's comments, resulting in an overall improvement in quality. Therefore, I recommend its publication in the current form.